# Ivermectin Inhibits HBV Entry into the Nucleus by Suppressing KPNA2

**DOI:** 10.3390/v14112468

**Published:** 2022-11-08

**Authors:** Anna Nakanishi, Hiroki Okumura, Tadahiro Hashita, Aya Yamashita, Yuka Nishimura, Chihiro Watanabe, Sakina Kamimura, Sanae Hayashi, Shuko Murakami, Kyoko Ito, Takahiro Iwao, Akari Ikeda, Tomoyasu Hirose, Toshiaki Sunazuka, Yasuhito Tanaka, Tamihide Matsunaga

**Affiliations:** 1Department of Clinical Pharmacy, Graduate School of Pharmaceutical Sciences, Nagoya City University, Nagoya 467-8603, Japan; 2Educational Research Center for Clinical Pharmacy, Faculty of Pharmaceutical Sciences, Nagoya City University, Nagoya 467-8603, Japan; 3Department of Virology and Liver Unit, Graduate School of Medical Sciences, Nagoya City University, Nagoya 467-8601, Japan; 4Department of Gastroenterology and Hepatology, Kumamoto University, Kumamoto 860-8556, Japan; 5Ōmura Satoshi Memorial Institute, Graduate School of Infection Control Sciences, Kitasato University, Tokyo 108-8641, Japan

**Keywords:** hepatitis B virus, ivermectin, importin α/β, karyopherin α, hepatitis B surface antigen, hepatitis B core protein, nuclear transfer, nuclear localization signal

## Abstract

Hepatitis B virus (HBV) specifically infects human hepatocytes and increases the risks of cirrhosis and liver cancer. Currently, nucleic acid analogs are the main therapeutics for chronic hepatitis caused by HBV infection. Although nucleic acid analogs can eliminate HBV DNA by inhibiting HBV reverse transcriptase, they cannot lead to negative conversion of covalently closed circular DNA (cccDNA) and hepatitis B surface antigen (HBsAg). In this study, we revealed that the antifilarial drug ivermectin suppresses HBV production by a different mechanism from the nucleic acid analog entecavir or Na^+^ taurocholate co-transporting polypeptide-mediated entry inhibitor cyclosporin A. Ivermectin reduced the levels of several HBV markers, including HBsAg, in HBV-infected human hepatocellular carcinoma cells (HepG2-hNTCP-C4 cells) and humanized mouse hepatocytes (PXB hepatocytes). In addition, ivermectin significantly decreased the expression of HBV core protein and the nuclear transporter karyopherin α2 (KPNA2) in the nuclei of HepG2-hNTCP-C4 cells. Furthermore, depletion of KPNA1–6 suppressed the production of cccDNA. These results suggest that KPNA1–6 is involved in the nuclear import of HBV and that ivermectin suppresses the nuclear import of HBV by inhibiting KPNA2. This study demonstrates the potential of ivermectin as a novel treatment for hepatitis B.

## 1. Introduction

The World Health Organization estimates that 296 million people globally were living with hepatitis B virus (HBV) infection in 2019, with 1.5 million new infections occurring each year [1]. Some cases of HBV infection progress to chronic hepatitis, which increases the risks of cirrhosis and liver cancer [2,3,4]. More than 800,000 people die each year of hepatitis B-related diseases, and, thus, treatment for hepatitis B is an important global issue [5]. The goal of treatment for patients with persistent HBV infection is to prevent progression to chronic liver failure and hepatocellular carcinoma by suppressing hepatitis and liver fibrosis [6,7,8]. Recently, the association between hepatitis B surface antigen (HBsAg) levels and carcinogenesis has been highlighted because the rates of progression to liver lesions and carcinogenesis were reported to be higher in patients with high HBsAg levels even though their HBV DNA levels were low [9]. Therefore, negative conversion of HBsAg is important for treatment success.

To treat HBV-infected patients, nucleic acid analogs and peg-interferon (PEG-IFN) have been used [10]. However, the probability of HBV elimination from cells is extremely low because the existing therapies cannot target covalently closed circular DNA (cccDNA) and integrated DNA, which are sources of HBsAg production [11,12]. Furthermore, although nucleic acid analogs can suppress HBV DNA production, long-term administration is required because relapse occurs following drug discontinuation [13]. Meanwhile, long-term nucleic acid analog administration is associated with the emergence of drug-resistant HBV mutants [14]. By contrast, PEG-IFN can induce HBsAg loss after prolonged treatment in responders [15,16,17]. However, the response rate is low, and severe side effects such as depression arise. For these reasons, novel therapeutic drugs such as HBV entry and capsid assembly inhibitors with different mechanisms from existing therapeutic drugs are needed [18,19,20].

HBV consists of genomic DNA stored in nucleocapsids and envelope antigens covering nucleocapsids [21,22]. HBV enters hepatocytes via Na^+^ taurocholate co-transporting polypeptide (NTCP), and then nucleocapsids are released from the envelope [23]. HBV nucleocapsids accumulate in the nucleus after entry into hepatocytes, and HBV genomic DNA released from nucleocapsids is converted to cccDNA by various enzymes in the host hepatocytes [24]. In addition, HBV levels are amplified through the production of various viral proteins and genomes using cccDNA as a template [25]. In the process from HBV invasion to cccDNA generation, the nuclear transfer of nucleocapsids has been suggested to involve the nuclear–cytoplasmic transporter importin [26]. Importins, also known as karyopherins, feature α and β subunits [27], and importin α is classified into seven subtypes (karyopherin α1–7 (KPNA1–7)) with different substrate specificities [28]. Importin α recognizes the specific amino acid sequence called the nuclear localization signal (NLS) on the substrate, allowing it to bind to the substrate [29]. Thereafter, the tripartite complex with importin β (karyopherin β1 (KPNB1)) is formed and then is transported into the nucleus [30]. The HBV nucleocapsid, which contains this NLS sequence [31], has been reported to be transported into the nucleus by importin α/β [32]. However, the subtype of importin α that transfers HBV into the nucleus is unclear [33,34].

Ivermectin is prepared by chemical modification of avermectin, which was isolated from actinomycetes by Ōmura et al. [35,36], and it exerts anthelmintic effects by inhibiting glutamate-gated chloride channels [37,38]. Therefore, ivermectin is used as a therapeutic drug for strongyloidiasis [39], scabies [40], and onchocerciasis caused by nematodes, mites, and filarial infection, respectively. Furthermore, ivermectin has also been used to treat onchocerciasis infection in tropical regions [41]. Recently, ivermectin was reported to inhibit human immunodeficiency virus type 1 (HIV-1) and dengue virus infection by specifically inhibiting the nuclear transport pathway involving importin α/β [42].

Therefore, this study investigated whether ivermectin can inhibit HBV infection and suppress the production of viral proteins such as HBsAg by inhibiting the nuclear transport pathway involving importin α/β.

## 2. Materials and Methods

### 2.1. Compounds

Ivermectin and importazole were purchased from Sigma-Aldrich (St. Louis, MO, USA), and entecavir (ETV) and cyclosporin A (CsA) were purchased from Toronto Research Chemicals (Toronto, Canada) and Nakalai Tesque (Kyoto, Japan), respectively.

### 2.2. Cell Culture

HepG2-hNTCP-C4, Hep38.7-Tet, and HepG2.2.15.7 cells were kindly provided by Dr. Koichi Watashi from the National Institute of Infectious Diseases [43,44]. HepG2-hNTCP-C4 cells with high susceptibility to HBV infection were maintained in Dulbecco’s Modified Eagle Medium/Nutrient Mixture F-12 (DMEM/F12; Wako, Osaka, Japan) supplemented with 10% fetal bovine serum (FBS; Sigma-Aldrich), 1% GlutaMAX Supplement (Thermo Fisher Scientific, Waltham, MA, USA), 10 mM HEPES (Sigma-Aldrich), 100 IU/mL penicillin G, 100 μg/mL streptomycin, 5 µg/mL insulin (Wako), and 1 mg/mL G418 (Nacalai Tesque) at 37 °C in a 5% CO_2_ incubator.

Hep38.7-Tet and HepG2.2.15.7 cells, which are HBV-replicating cells, were cultured in DMEM/F12, GlutaMAX supplement (Thermo Fisher Scientific) supplemented with 10% FBS, 10 mM HEPES, 100 IU/mL penicillin G, 100 μg/mL streptomycin, 5 μg/mL insulin, 50 μM hydrocortisone (Sigma-Aldrich), 400 μg/mL G418, and 0.5 μg/mL tetracycline (Sigma-Aldrich) at 37 °C in a 5% CO_2_ incubator. In the HBV replication experiment, Hep38.7-Tet cells were exposed to 10 µM ivermectin, 1 µM ETV, or 0.1% dimethyl sulfoxide (DMSO) as a vehicle control for 4 days without G418 and tetracycline.

PXB hepatocytes, which were fresh hepatocytes collected from PXB mice with >90% human hepatocytes, were purchased from Phoenix Bio Co., Ltd. (Hiroshima, Japan) and cultured in DMEM low glucose (Thermo Fisher Scientific) supplemented with 10% FBS, 20 mM HEPES, 100 IU/mL penicillin G, 100 μg/mL streptomycin, 15 μg/mL l-proline (Sigma-Aldrich), 0.25 μg/mL insulin, 50 nM dexamethasone (Sigma-Aldrich), 5 ng/mL epidermal growth factor (Peprotech, Cranbury, NJ, USA), 0.1 mM ascorbic acid 2-phosphate (Wako), and 2% DMSO (Nacalai Tesque).

### 2.3. Infection of HepG2-hNTCP-C4 Cells with HBV

HepG2-hNTCP-C4 cells were seeded onto plates coated with 0.01% collagen type I (Research Institute for the Functional Peptides Co., Yamagata, Japan) at a density of 1 × 10^5^ cells/cm^2^. After 2 days, the medium was replaced with fresh medium containing 3% DMSO in the absence of G418 with or without ivermectin (10 μM) or importazole (10 μM), and the cells were incubated at 37 °C in a 5% CO_2_ incubator. After 24 h, HepG2-hNTCP-C4 cells were infected with HBV purified from the concentrated culture supernatant of HepG2.2.15.7 cells. The supernatant was concentrated to 4.9 × 10^9^ copies/mL using filtered centrifuge tubes (Amicon^®^ Ultra-15 100K, Merck Millipore Ltd., Burlington, MA, USA). HepG2-hNTCP-C4 cells were infected with 1 × 10^3^ copies/cell HBV using medium containing 3% DMSO and 4% polyethylene glycol (PEG)-8000 (Sigma-Aldrich) in cells. In the treatment groups, ivermectin or importazole was added to the medium. At 24 h after HBV infection, the cells were washed three times with phosphate-buffered saline (PBS), and the medium was replaced with fresh medium containing 3% DMSO. After 48 h, the cells were again washed three times with PBS, and the medium was replaced with fresh medium without DMSO. Then, the new medium was changed 3 and 6 days after infection, and culture supernatant was collected for extracellular HBV DNA and HBsAg measurements. At 9 days post-infection (dpi), HepG2-hNTCP-C4 cells and culture supernatant were collected for subsequent analyses.

### 2.4. Infection of PXB Hepatocytes with HBV

PXB hepatocytes were infected with 10 copies/cell HBV genotype C purified from PXB mouse serum in medium containing 2% DMSO and 4% PEG-8000. At 24 and 48 h after HBV infection, the cells were washed three times with PBS, and the medium was replaced with fresh medium containing 2% DMSO. Ivermectin (10 μM) was added from one day prior to infection until the day after infection. The new medium was changed at 3 and 8 dpi, and the culture supernatant was collected for extracellular HBV DNA and HBsAg measurements. At 13 dpi, PXB hepatocytes and culture supernatant were collected for subsequent analyses.

### 2.5. Cytotoxicity Assay

Cell viability was measured using Cell Counting Kit-8 (CCK-8; Dojindo Laboratories, Kumamoto, Japan) according to the manufacturer’s instructions. Briefly, after collecting the culture supernatant of HepG2-hNTCP-C4 cells, the medium was replaced with medium containing 10-fold–diluted CCK-8 solution. The cells were incubated at 37 °C for 45 min, and the absorbance at 450 nm was measured using a microplate reader. Cell viability was calculated using the control as 100%.

### 2.6. HBV DNA Quantification by Quantitative Polymerase Chain Reaction (qPCR)

HBV DNA was extracted using SMITEST EX-R&D Kits (Medical & Biological Laboratories Co., Ltd., Tokyo, Japan) from the culture supernatants of Hep38.7-Tet cells and HBV-infected HepG2-hNTCP-C4 cells and PXB hepatocytes. HBV DNA was quantified by qPCR using the StepOnePlus Real-Time PCR System (Thermo Fisher Scientific). The primers and probes for HBV DNA, which were located in the S gene, were used as previously reported [45,46], and they consisted of the forward primer 5′-CAC ATC AGG ATT CCT AGG ACC-3′, the reverse primer 5′-AGG TTG GTG AGT GAT TGG AG-3′, and the TaqMan probe 5′-(FAM)-CAG AGT CTA GAC TCG TGG TGG ACT TC-[6-carboxytetramethylrhodamine (TAMRA)]-3′. qPCR was performed by DNA polymerase activation at 95 °C for 10 min, followed by 45 cycles of denaturation (95 °C for 15 s) and annealing and elongation reaction (65 °C for 1 min) using EagleTaq Universal Master Mix (ROX) (F. Hoffmann-La Roche, Ltd., Basel, Switzerland).

### 2.7. cccDNA Quantification by Droplet Digital Polymerase Chain Reaction (ddPCR)

DNA were extracted using SMITEST EX-R&D Kits according to the manufacturer’s instructions. To increase the specificity for HBV cccDNA in ddPCR, 1 μg of cell DNA was treated with Plasmid-Safe ATP-Dependent DNase (Lucigen Corporation, Middleton, WI, USA) to degrade HBV rcDNA [47,48]. The reaction was conducted via enzymatic treatment at 37 °C for 30 min, followed by inactivation at 70 °C for 30 min. Droplets including 20 ng of DNA per assay were generated using ddPCR Supermix and Droplet Generation Oil with the QX200 Droplet Digital PCR System (Bio-Rad Laboratories, Inc., Hercules, CA, USA). After droplet generation, cccDNAs were amplified using the forward primer 5′-ACG GGG CGC ACC TCT CTT TAC GCG G-3′, the reverse primer 5′-CAA GGC ACA GCT TGG AGG CTT GAA C-3′, and the probe 5′-(FAM)-AAC GAC CGA CCT TGA GGC AT-(MGB)-3′ with a C1000 touch Thermal Cycler (Bio-Rad Laboratories, Inc.). cccDNA levels were normalized by the number of cells. The number of cells was calculated by measuring the copy number of the human ribonuclease P/MRP subunit p30 (hRPP30) (Bio-Rad Laboratories, Inc.) [49]. hRPP30 copy numbers were separately determined using DNA that was not treated with Plasmid-Safe ATP-Dependent DNase.

### 2.8. Measurement of HBsAg and Hepatitis B Core-Related Antigen (HBcrAg) Levels in Culture Supernatants

The levels of HBsAg and HBcrAg in the culture supernatant were measured using the Lumipulse G 1200 fully automated chemiluminescent enzyme immunoassay system (Fujirebio, Tokyo, Japan).

### 2.9. PreS1 Binding Assay

A preS1 peptide consisting of aa 2–48 of the HBV preS1 region with amino-terminal myristoylation, as well as carboxy-terminal TAMRA-conjugated derivatives thereof (preS1-TAMRA; Myr-GTNLSVPNPL GFFPDHQLDP AFGANSNNPD WDFNPNKDHW PEANQV-C(TAMRA)), was synthesized by Scrum (Tokyo, Japan). To examine whether ivermectin inhibits the NTCP-mediated entry of HBV into cells, HepG2-hNTCP-C4 cells were treated with ivermectin (10 μM), CsA (10 μM), or 0.1% DMSO (vehicle control) for 3 h and 40 nM preS1-TAMRA for the last 30 min. Then, the cells were washed three times with PBS to remove free peptides, fixed with 4% paraformaldehyde for 30 min at room temperature, and finally stained with 4′,6-diamidino-2-phenylindole (DAPI; Dojindo Laboratories) as previously described [50]. Fluorescence observation and quantification of positive cells were performed using the Operetta High-Content Imaging System (PerkinElmer, Waltham, MA, USA).

### 2.10. Hepatitis B Core Protein (HBc) Plasmid Transfection

A plasmid expressing HBc [51] was constructed by inserting the sequence (1901–2449) encoding the preC/C protein of HBV DNA clone D_IND60 (GenBank accession no. AB246347) [52] between the *Eco*RI and *Bst*BI restriction sites of pcDNA3.1 (+)/myc-His C (Thermo Fisher Scientific).

HepG2-hNTCP-C4 cells were seeded onto collagen-coated 96-well plates at 1.5 × 10^4^ cells/well for immunofluorescence staining or collagen-coated 6-well plates at 4 × 10^5^ cells/well for the Simple Western assay and incubated for 24 h at 37 °C in a 5% CO_2_ incubator. Then, ivermectin (5 μM) or 0.1% DMSO (vehicle control) was added for 24 h. To prepare for transfection of the plasmid expressing HBc (pCMV-HBc) into HepG2-hNTCP-C4 cells, Lipofectamine 3000 (Thermo Fisher Scientific), P3000 reagent, and pCMV-HBc (100 ng/100 µL medium) were mixed in OPTI-MEM (Thermo Fisher Scientific) and incubated for 12 min at room temperature. Then, the cells were transfected with pCMV-HBc in the HepG2-hNTCP-C4 medium containing 1% FBS without penicillin–streptomycin solution for 1 h. At 1 h and 24 h after transfection, the medium was replaced with HepG2-hNTCP-C4 medium containing 10% FBS without penicillin–streptomycin solution and supplemented with ivermectin (5 µM) or 0.1% DMSO. At 48 h after transfection, the cells were fixed with 4% paraformaldehyde for 30 min at room temperature.

### 2.11. Small Interfering RNA (siRNA) Transfection

siRNA was transfected used Lipofectamine RNAiMAX Transfection Reagent (Thermo Fisher Scientific) at the time of cell seeding (3 days before infection) in a 48-well plate. Lipofectamine RNAiMAX Transfection Reagent containing siRNA or negative control (20 nM) was added to OPTI-MEM, followed by incubation at room temperature for 20 min. The following Silencer Select pre-designed siRNAs were used: KPNA1 (s223980), KPNA2 (s7920), KPNA3 (s7924), KPNA4 (s7927), KPNA5 (s7931), KPNA6 (s24243), KPNB1 (s7917), and negative control #1 (all from Ambion). The cells were transfected by replacing the medium with HepG2-hNTCP-C4 medium containing each siRNA and 10% FBS without penicillin–streptomycin solution. After 2 days, the medium was replaced with fresh HepG2-hNTCP-C4 medium containing 3% DMSO, and cells were incubated for 24 h at 37 °C in a 5% CO_2_ incubator. The transfected cells were then infected with HBV according to the protocol described in Section 2.3. The cells were collected to confirm the specific depletion of each gene at 2 dpi, and then cells and culture supernatant were collected for HBV marker assay at 9 dpi.

### 2.12. Reverse Transcription Real-Time Quantitative PCR

Total RNA was extracted using the Agencourt RNAdvance Tissue Kit (Beckman Coulter Inc., Brea, CA, USA) according to the manufacturer’s instructions. The cDNA was obtained via reverse transcription of 500 ng total RNA using the ReverTra Ace qPCR RT Kit (TOYOBO, Osaka, Japan). The cDNA level was analyzed by real-time qPCR using a KAPA PROBE Fast qPCR Kit (KAPA Biosystems, Wilmington, MA, USA) and the LightCycler 96 system (Roche). The primers and probes are listed in Table 1. The mRNA levels were normalized to that of the human hypoxanthine phosphoribosyl transferase (*HPRT*) gene.

### 2.13. Immunofluorescence Staining

The method of immunofluorescence staining was modified from the protocol of Okumura et al. [53]. As the primary and secondary antibodies, anti-HBc antibody (1:250, Institute of Immunology; 2Z19C18, Tokyo, Japan) and Alexa Fluor 488-conjugated secondary antibody (1:200, Thermo Fisher Scientific), respectively, were diluted in PBS with 1% bovine serum albumin. Fluorescence was observed using a BZ-X810 all-in-one fluorescence microscope (Keyence, Osaka, Japan) or Operetta High-Content Imaging System. The ratio of HBc nuclear transfer in pCMV-HBc–transfected cells was calculated using ImageJ (US National Institutes of Health, Bethesda, MD, USA). The rate of HBc nuclear transfer was calculated using the following formula:(1)$$\mathrm{HBc}\;\mathrm{nuclear}\;\mathrm{transfer}\;\left(\%\right)=\frac{\mathrm{HBc}\;\mathrm{in}\;\mathrm{nuclear}\;\mathrm{region}}{\mathrm{Total}\;\mathrm{HBc}}\times 100$$

The positive region of DAPI was defined as the nuclear region.

### 2.14. Western Blotting and Simple Western Assay

Nuclear and cytoplasmic proteins were extracted using a LysoPure Nuclear and Cytoplasmic Extractor Kit (Wako) or Nuclear Complex Co-IP Kit (Active Motif, Carlsbad, CA, USA) based on the manufacturer’s instructions. The Western blotting method was modified from that of Okumura et al. [53]. Briefly, the extracted proteins were separated using 10% sodium dodecyl sulfate–polyacrylamide gel electrophoresis and transferred onto a polyvinylidene difluoride membrane. The membranes were blocked with BlockAce (KAC Co., Kyoto, Japan) for 60 min at room temperature and then incubated with the primary antibodies presented in Table 2 at 4 °C overnight. The membranes were incubated with the peroxidase-conjugated secondary antibody (1:1000 for KPNA2, 1:15,000 for others) for 60 min at room temperature. Then, the protein bands were analyzed using Pierce ECL Western Blotting Substrate (Thermo Fisher Scientific) for KPNA2 and SignalFire Elite ECL Reagent (Cell Signaling Technology, Danvers, MA, USA) for other proteins and visualized using an Amersham Imager 600 (GE Healthcare, Chicago, IL, USA).

For the Simple Western assay, the total protein concentration was measured according to the manual of the TaKaRa BCA Protein Assay Kit (Takara Bio, Shiga, Japan). The data were analyzed using the WES system and 12–230 kDa Separation 8 × 25 Capillary Cartridges (ProteinSimple, San Jose, CA, USA) according to the manufacturer’s instructions. All samples were diluted with 0.1× Sample Buffer to 0.2 mg/mL total protein. After separation of proteins by molecular weight, target proteins were identified using anti-HBc primary antibody (1:50), anti-Lamin B1 primary antibody (1:50, Proteintech; 12987-1-AP, Rosemont, IL, USA), horseradish peroxidase-conjugated secondary antibody, the Anti-mouse detection module, and the Anti-rabbit detection module (undiluted, ProteinSimple) and quantified by a chemiluminescence method. The total protein content of each sample was then measured by the same device using the Total Protein Detection Module (ProteinSimple) and used for correction. The results were analyzed using Compass software for Simple Western (ProteinSimple).

### 2.15. Statistical Analysis

Each experiment was performed at least twice, and all data are expressed as the mean ± SD. Statistical comparisons were performed using a two-tailed Student’s *t*-test for between two groups and one-way analysis of variance (ANOVA) with a two-tailed Dunnett’s test for three or more groups, respectively. Statistical comparisons of the time course data were performed using two-way repeated-measures ANOVA for multiple comparisons. In the siRNA knockdown experiments, only the group transfected with siRNA was subjected to statistical analysis. Statistical analyses were performed using SPSS Statistics software version 25.0 (IBM Japan, Tokyo, Japan). The results were considered statistically significant at *p* < 0.05.

## 3. Results

### 3.1. Ivermectin Inhibits HBV Infection in HepG2-hNTCP-C4 Cells

We examined the effect of ivermectin on HBV infection in HepG2-hNTCP-C4 cells using HBV markers. The cells were exposed to ivermectin (Figure 1A) for 48 h starting 1 day prior to HBV infection and analyzed for various HBV markers and cell viability (Figure 1B). The levels of both HBV DNA and HBsAg in the culture supernatant were significantly reduced by ivermectin exposure in a time-dependent manner (Figure 1C). At 9 dpi, ivermectin reduced HBcrAg and cccDNA levels to 54% and 61% of the control, respectively (Figure 1D,E), and concentration-dependently reduced HBsAg levels (Figure 1F). Furthermore, no cytotoxicity was observed after ivermectin treatment (Figure 1G).

### 3.2. Anti-HBV Effects of Ivermectin in the Initial HBV Infection in PXB Hepatocytes

We used PXB hepatocytes to confirm whether the anti-HBV effects of ivermectin were also observed in normal human hepatocytes. Similarly as presented in Figure 1, exposure of PXB hepatocytes to ivermectin for 48 h starting 1 day prior to HBV infection resulted in time-dependent decreases of both HBV DNA and HBsAg levels in the culture supernatant and significantly reduced levels of HBcrAg and cccDNA at 13 dpi (Figure 2A–E). Additionally, no ivermectin-induced cytotoxicity was observed in PXB hepatocytes (Figure 2F). These results suggest that ivermectin exerts anti-HBV effects in both hepatocellular carcinoma cells and normal human hepatocytes.

### 3.3. Ivermectin Inhibits HBV Infection via a Novel Mechanism

We investigated whether ivermectin exhibits previously described mechanisms such as inhibition of reverse transcriptase and NCTP-mediated HBV entry. Exposure of Hep38.7-Tet cells to ETV for 4 days in the absence of tetracycline significantly decreased the production of HBV DNA compared to the control findings (Figure 3A). However, exposure to ivermectin had no effect on HBV DNA production. Next, we investigated whether ivermectin inhibits NCTP-mediated HBV entry using the preS1 binding assay with ivermectin or CsA. CsA binds to NTCP and inhibits the interaction between NTCP and the PreS1 region of the HBV envelope protein, thereby suppressing HBV entry into hepatocytes [54]. Treatment with CsA for 3 h significantly inhibited the entry of preS1 protein into HepG2-hNTCP-C4 cells (Figure 3B). However, ivermectin induced the accumulation of preS1 protein in the cells to the same extent as the control. Additionally, we performed a preS1 internalization assay to investigate whether ivermectin affected not only the binding of HBV to NTCP but also the subsequent uptake (Appendix A). In the preS1 internalization assay, preS1-TAMRA was allowed to attach to the cell surface at 4 °C (4 °C step), followed by intracellular uptake of preS1-TAMRA at 37 °C (37 °C step). When CsA was added from the 4 °C step, it inhibited preS1 binding, resulting in reduced cellular uptake of preS1-TAMRA (Appendix A). Alternatively, ivermectin showed the same degree of intracellular uptake of preS1-TAMRA as the control when added from either the 4 °C or 37 °C step (Appendix A). These results suggest that ivermectin inhibits HBV production by a different mechanism other than the inhibition of reverse transcriptase and HBV entry into hepatocytes.

### 3.4. Ivermectin Inhibits the Nuclear Transfer of HBc

We used pCMV-HBc–transfected HepG2-hNTCP-C4 cells to investigate whether the subcellular localization of HBc, which plays an important role in the nuclear transfer of HBV, is changed by ivermectin treatment. Nuclear and cytoplasmic proteins were properly fractionated and extracted (Appendix A). Immunostaining illustrated that the nuclear localization of HBc was significantly inhibited by ivermectin compared to the effects of the control (Figure 4A and Appendix A). We also analyzed the amount of HBc protein in the nuclear and cytoplasmic fractions using the Simple Western assay. The results demonstrated that ivermectin significantly reduced HBc content in the nuclear fraction (Figure 4B). Thus, these results suggested that ivermectin suppresses HBV production by inhibiting the nuclear transfer of HBc.

### 3.5. Ivermectin Decreases the Nuclear Localization of KPNA2

To date, seven subtypes of importin α have been found in humans, and they are designated KPNA1–7 [33,34]. However, the expression of each subtype in human hepatocytes and the subtypes that are inhibited by ivermectin are unclear. Therefore, we investigated the expression of KPNA1–7 and KPNB1 in HepG2-hNTCP-C4 cells and PXB hepatocytes and the effect of ivermectin on importin α/β expression. Although the gene expression pattern differed between HepG2-hNTCP-C4 cells and PXB hepatocytes, KPNA1–6 and KPNB1 were expressed in both cell lines (Figure 5A). In particular, KPNA2, KPNA6, and KPNB1 were more highly expressed in HepG2-hNTCP-C4 cells than in PXB hepatocytes. Conversely, KPNA7 was hardly expressed in either cell line.

To investigate the interaction between ivermectin and importin, the protein expression of KPNA1–6 and KPNB1 in the nucleus was measured following treatment with ivermectin. Nuclear and cytoplasmic proteins were properly fractionated and extracted (Appendix A). The results indicated that nuclear KPNA2 expression was significantly decreased by ivermectin (Figure 5B). However, no significant difference was shown in the nuclear expression of KPNA subtypes other than KPNA2. These results suggest that ivermectin inhibits the intranuclear localization of KPNA2.

### 3.6. Multiple Subtypes of Importin α Are Involved in HBV Infection

To determine which subtypes of importin are involved in HBV infection, we infected cells with HBV after depleting each importin. The specific depletion of targeted importin genes (KPNA1–6 and KPNB1) by siRNAs was confirmed (Appendix A). No cytotoxicity was observed with KPNA1–6 depletion, while cell death was observed with KPNB1 depletion as the number of days after siRNA transfection increased (Appendix A). Depletion of each KPNA subtype significantly reduced the production of cccDNA compared to the findings in the control group (Figure 6A). In addition, depletion of KPNA1, KPNA2, KPNA4, and KPNA6 significantly reduced HBsAg levels in the culture supernatant compared to the control findings (Figure 6B). However, depletion of KPNA3 increased HBsAg levels (Figure 6B). The depletion of KPNB1 led to a significant decrease in HBsAg levels due to cell death, while cccDNA levels were significantly increased (Figure 6A,B). To further investigate KPNB1 and HBV infection, HepG2-hNTCP-C4 cells infected with HBV were exposed to importazole, a specific inhibitor of KPNB1. However, HBsAg levels did not decrease (Figure 6C). These results suggest that KPNA, but not KPNB1, is mainly involved in HBV infection, and the suppression of KPNA1, KPNA2, KPNA4, and KPNA6 also inhibits HBV replication.

## 4. Discussion

This is the first report to show that ivermectin suppresses HBV infection and that ivermectin may inhibit the nuclear transfer of HBc by inhibiting importin α/β. Ivermectin was previously reported to suppress RNA viruses, such as HIV and dengue virus, and DNA viruses such as pseudorabies virus [42,55]. The mechanism by which ivermectin suppresses these viruses is inhibiting host-derived importin α/β complexes, thereby preventing the entry of viruses carrying NLS into the nucleus [42]. In this study, we found that ivermectin suppressed HBV infection in vitro, and the suppressive effects were mainly based on the inhibition of importin α1 (i.e., KPNA2).

The study results suggested that ivermectin can inhibit HBV production by suppressing the accumulation of cccDNA (Figure 1C–F), and further analysis illustrated that the effects were not linked to the inhibition of cell proliferation (Figure 1G). The results using PXB hepatocytes demonstrated that ivermectin exerts anti-HBV effects on both human hepatocellular carcinoma cells and normal human hepatocytes and does not cause cytotoxicity (Figure 2). In addition, unlike ETV and CsA, the mechanism of anti-HBV effects by ivermectin was unrelated to the inhibition of HBV DNA replication and NTCP-mediated entry into hepatocytes (Figure 3 and Appendix A). These results indicate that ivermectin inhibits HBV replication by a novel mechanism differing from those of ETV and CsA. Alternatively, there was no significant difference in HBV DNA and HBsAg levels between the control and ivermectin treatment at 3 dpi (Figure 1C and Figure 2B,C). It has been reported that cccDNA is gradually produced over 3 days following infection [56]. The lack of significant differences at 3 dpi may be because the newly produced HBV markers in the culture supernatant were below the limit of detection rather than the weak inhibition of HBc nuclear entry by ivermectin in the very early stages of infection.

The nucleocapsid constituted by the HBc protein is involved in the nuclear transfer of HBV. Importin α/β has been reported to participate in the nuclear entry of HBc [32], and ivermectin has been reported to inhibit importin α/β [42]. Since the nuclear accumulation of HBc was significantly reduced by ivermectin in this study (Figure 4), it was predicted that ivermectin inhibits nuclear transfer of HBc by importin α/β. However, the HBc applied in this study did not contain the HBV genome and may exhibit different intracellular behavior from that of the original HBV. Therefore, it is also necessary to investigate whether ivermectin inhibits the nuclear transfer of HBc in systems containing the HBV genome.

Then, we investigated the interaction between ivermectin and importin in hepatocytes and the association between importin and HBV infection. Ivermectin significantly inhibited the accumulation of KPNA2 in the nuclei of HepG2-hNTCP-C4 cells (Figure 5B). Contrarily, ivermectin exposure had no effect on the expression of KPNA6 even though it was the second most abundantly expressed importin α subtype in HepG2-hNTCP-C4 cells (Figure 5). Therefore, ivermectin can inhibit the intranuclear localization of KPNA2, irrespective of the expression levels of KPNA subtypes. Because we only analyzed the expression of nuclear importin, the possibility is undeniable that ivermectin binds to importin instead of HBc and enters the nucleus. The mechanism of action of ivermectin has been reported to be through binding to the NLS-binding pocket of importin α [42]. Therefore, importins, except KPNA2, whose nuclear expression was not significantly different in this study, may be also involved in inhibiting HBV nuclear entry by ivermectin. Further investigations using AlphaScreen assay or a chemical protein knockdown technique using specific and nongenetic inhibitor of apoptosis protein-dependent protein erasers (SNIPERs) must be undertaken to determine whether ivermectin decreases the binding of importins and intracytoplasmic HBc.

In the study, the depletion of each KPNA gene led to significantly reduced cccDNA production, suggesting the involvement of KPNAs in the HBV infection process up to cccDNA production, including the nuclear transport of the HBV nucleocapsid (Figure 6A). Although we have not examined the effects of KPNA1–6 and KPNB1 depletion on HBc nuclear transfer yet, this would provide a more detailed insight regarding the reduced levels of cccDNA. By contrast, HBsAg production was significantly reduced by the depletion of KPNA1, KPNA2, KPNA4, and KPNA6, whereas KPNA3 depletion resulted in significantly increased HBsAg production despite the significant decline in cccDNA production (Figure 6A,B). Although cccDNA production is likely dependent on the nuclear transport of HBc, the production of HBsAg depends on cccDNA production and subsequent HBV replication mechanisms such as transcription [57]. KPNA3 is involved in the activation of NF-κB signaling, and KPNA3 depletion was reported to promote viral replication via the suppression of antiviral gene expression [58]. Therefore, in this study, we assumed that the same mechanism was involved in the increased production of HBsAg. KPNA6 depletion more strongly suppressed cccDNA and HBsAg production than depletion of the other KPNA subtypes (Figure 6A,B). KPNA6 has been identified as a positive regulator of influenza virus replication [59]. Although the inhibitory effect of ivermectin on the nuclear import of KPNA6 was weak (Figure 5B), the anti-HBV effect might be increased by enhancing the binding specificity of ivermectin for KPNA6. Furthermore, no cytotoxicity was observed in cells with KPNA1–6 depletion (Appendix A). This may be because when one KPNA subtype does not function, the other subtypes compensate. In fact, knockdown of each KPNA subtype did not significantly suppress HBV more than ivermectin, and no significant difference was observed in comparison with non-transfected cells (Figure 6A,B). Previously, it was reported that HIV-1 gets transferred into the nucleus via multiple subtypes of KPNA [60]. Therefore, we considered that multiple KPNA subtypes are similarly involved in HBV infection, and that other KPNAs, except for the knockdown KPNA, may have complemented HBV nuclear transfer, thereby weakening the inhibitory effect of cccDNA production. The results in Figure 5 and Figure 6 suggest that ivermectin exerts its anti-HBV effect by inhibiting KPNA2, thereby suppressing the nuclear transport of HBc and factors involved in HBV replication. The overexpression of KPNA2 and confirmation of the attenuated effect of ivermectin may aid our understanding of this finding in more detail.

In addition, depletion of KPNB1 significantly reduced HBsAg production but significantly increased cccDNA production (Figure 6A,B). Cell death was only observed with KPNB1 depletion (Appendix A). Importin β plays a central role in the nuclear transport of signaling molecules such as transcription factors as a monomer or as a heterodimer with adaptors such as importin α [29]. In a previous study, KPNB1 depletion was reported to induce cell death by reducing the efficiency of KPNB1 transfer to the nucleus [61]. Therefore, cell survival might have been inhibited by KPNB1 depletion, resulting in a decrease in the amount of HBsAg in the culture supernatant. Furthermore, KPNB1 depletion increased the expression of several KPNA subtypes (Appendix A). Previous research speculated that importin α is singly transferred into the nucleus because the amount of importin β is limited by specific conditions, such as infection [62]. It is possible that KPNA expression was promoted to compensate for the depletion of KPNB1 and that KPNA promoted HBV infection, resulting in increased cccDNA levels and cytotoxicity. Alternatively, KPNB1 knockdown reportedly decreased cccDNA production from HepDES19 cells, which are HBV-replicating cells [63], and this was inconsistent with the results of our study. Hereby, we deduced that the differences in the properties of HepG2-hNTCP-C4 cells and HepDES19 cells may have resulted in increased cytotoxicity and HBV infection due to KPNB1 depletion. To further investigate the association between HBV infection and KPNB1, we analyzed the anti-HBV effect of importazole, which surprisingly failed to suppress HBV infection even at five-fold higher concentrations than required for ivermectin (Figure 6C and Appendix A). In our study, cccDNA production was not reduced by KPNB1 depletion and HBsAg production was not inhibited by importazole (Figure 6A,C), suggesting that HBV is transported into the nucleus without importin β.

## 5. Conclusions

Our results demonstrate that ivermectin reduced the production of HBV-related markers, such as cccDNA and HBsAg, by inhibiting KPNA2 in hepatocytes. This study signifies the novel potential of ivermectin for HBV therapy.

## Figures and Tables

**Figure 1 viruses-14-02468-f001:**
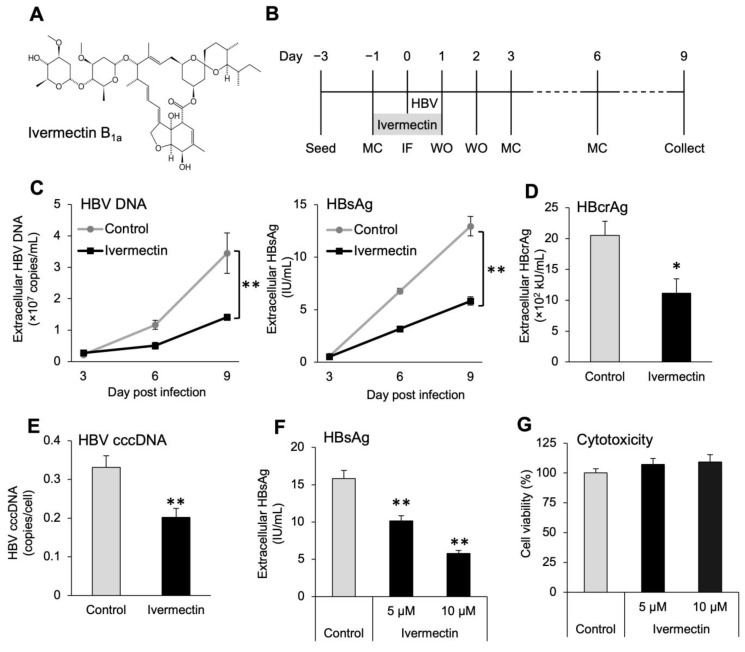
Ivermectin inhibited HBV infection in HepG2-hNTCP-C4 cells. (**A**) Chemical structure of ivermectin. (**B**) Experimental scheme for HBV infection in HepG2-hNTCP-C4 cells. HepG2-hNTCP-C4 cells were treated with 5 µM (**F**,**G**) or 10 μM (**C**–**G**) ivermectin for 48 h. MC, medium change; IF, infection; WO, washout with PBS. (**C**) Time-dependent changes in HBV DNA HBsAg levels in the culture supernatant. (**D**,**E**) Levels of HBcrAg in the culture supernatant and intracellular cccDNA levels after 9 days of infection. (**F**) Concentration-dependent effect of ivermectin on HBsAg levels after 9 days of infection. (**G**) Cell viability after 48 h of ivermectin treatment. The absorbance of the control was defined as 100%. Data are presented as the mean ± SD ((**C**,**D**,**F**), *n* = 4; (**E**), *n* = 6; (**G**), *n* = 5). * *p* < 0.05, ** *p* < 0.01.

**Figure 2 viruses-14-02468-f002:**
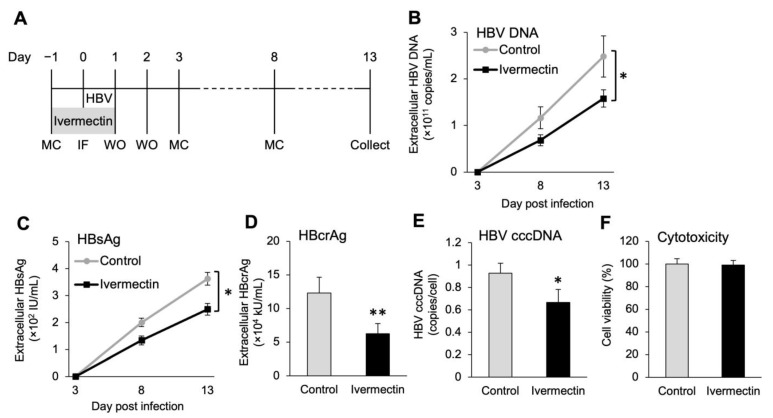
Ivermectin inhibited HBV production in PXB hepatocytes. (**A**) Experimental scheme for ivermectin exposure in the early stages of HBV infection. PXB hepatocytes were treated with 10 μM ivermectin for 48 h. MC, medium change; IF, infection, WO: washout with PBS. (**B**,**C**) Time-dependent changes in HBV DNA and HBsAg levels in the culture supernatant. (**D**,**E**) The levels of HBcrAg in the culture supernatant and intracellular cccDNA levels after 9 days of infection. (**F**) Cell viability after 13 days of HBV infection. The absorbance of the control was 100%. Data are presented as the mean ± SD ((**B**–**D**), *n* = 4; (**E**), *n* = 3). * *p* < 0.05, ** *p* < 0.01.

**Figure 3 viruses-14-02468-f003:**
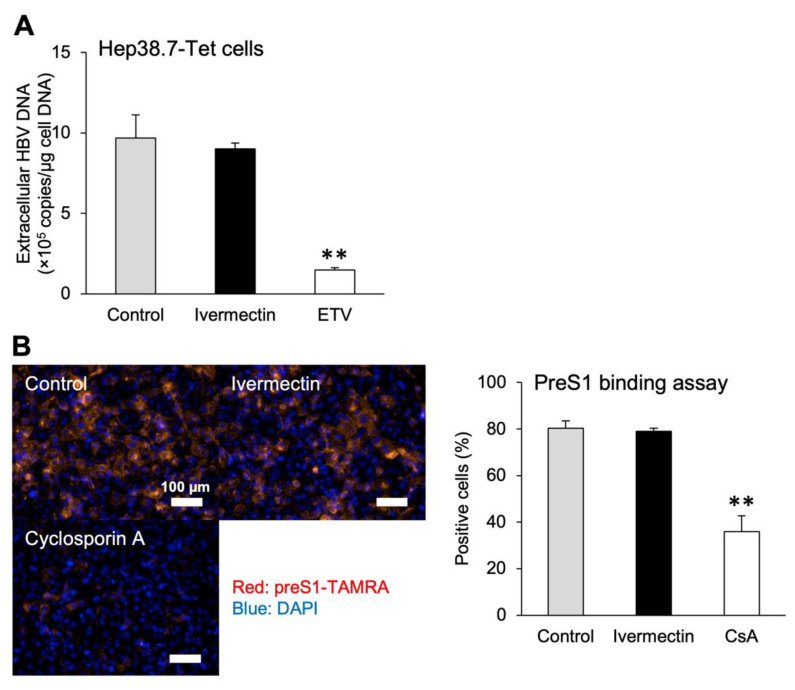
Ivermectin exerted its effects through a different mechanism than existing HBV-targeted compounds. (**A**) Extracellular HBV DNA levels in Hep38.7-Tet cells 4 days after tetracycline removal. Hep38.7-Tet cells were treated with 10 μM ivermectin, 1 μM ETV, or 0.1% DMSO (vehicle control) for 4 days. (**B**) Fluorescence imaging and quantification of preS1-TAMRA–positive cells in the PreS1 binding assay. HepG2-hNTCP-C4 cells were treated with 10 μM ivermectin, 10 μM CsA, or 0.1% DMSO (vehicle control) for 3 h, followed by reaction with 40 nM preS1-TAMRA for 30 min. Data are presented as the mean ± SD ((**A**), *n* = 3; (**B**), *n* = 4). ** *p* < 0.01.

**Figure 4 viruses-14-02468-f004:**
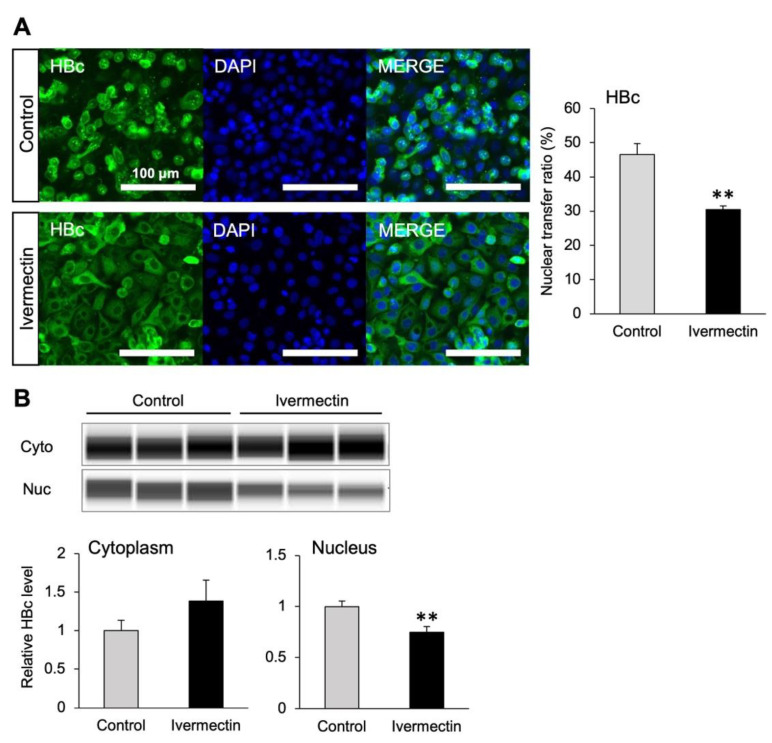
Ivermectin decreased HBc entry into the nucleus. (**A**) Distribution and percentage of HBC-positive cells in the nucleus 48 h after HBc transfection. HepG2-hNTCP-C4 cells were treated with 5 μM ivermectin or 0.1% DMSO (vehicle control) from 24 h before to 48 h after HBc transfection. (**B**) Quantification of HBc expression in the nucleus and cytoplasm of HepG2-hNTCP-C4 cells at 48 h after HBc transfection. HepG2-hNTCP-C4 cells were treated with 5 μM ivermectin or 0.1% DMSO (vehicle control) from 24 h before to 48 h after HBc transfection. Expression was corrected for the total protein levels of each sample. In the graphs, the protein expression of the control is defined as 1. Data are presented as the mean ± SD ((**A**), *n* = 4; (**B**), *n* = 3). ** *p* < 0.01.

**Figure 5 viruses-14-02468-f005:**
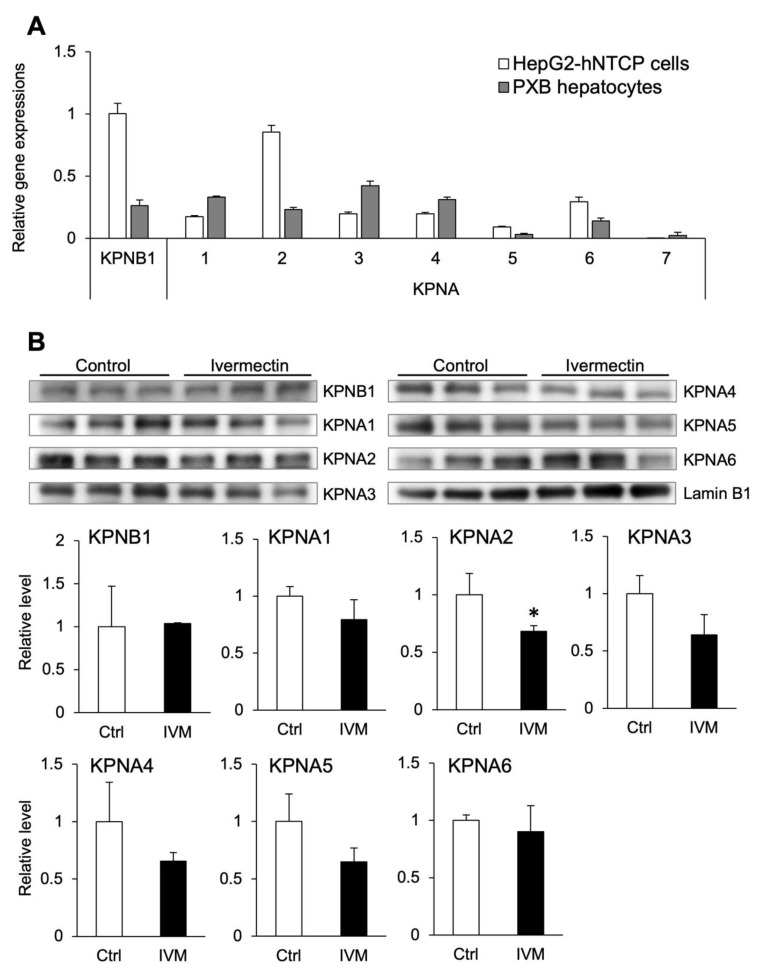
Ivermectin decreased the expression of importin α in the nucleus. (**A**) The gene expression of importin α/β in HepG2-hNTCP-C4 cells and PXB hepatocytes at 48 h after seeding. The genes for importin β and importin α family members were represented by karyopherin β1 (KPNB1) and karyopherin α1–7 (KPNA1–7), respectively. The gene expression of KPNB1 in HepG2-hNTCP-C4 cells was defined as 1. (**B**) Protein expression of nuclear KPNA1–6 and KPNB1 after treatment of HepG2-hNTCP-C4 cells with 10 μM ivermectin for 48 h. Expression was corrected for that of lamin B1 (a nuclear marker) for each sample. In the graphs, the protein expression in the vehicle control group (Ctrl; 3% DMSO) is defined as 1. IVM, ivermectin. Data are presented as the mean ± SD ((**A**) HepG2-hNTCP-C4 cells, *n* = 4; PXB hepatocytes, *n* = 3; (**B**), *n* = 3). * *p* < 0.05.

**Figure 6 viruses-14-02468-f006:**
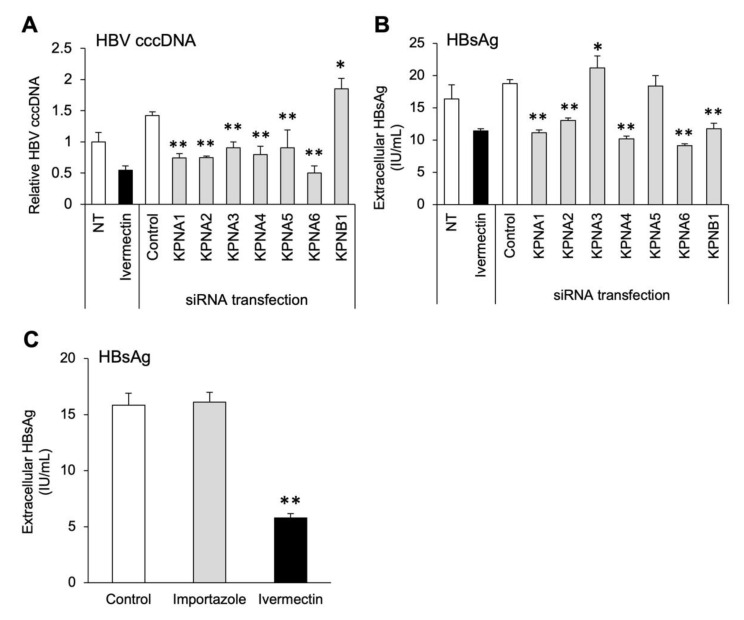
Multiple subtypes of importin α were involved in HBV infection. (**A**,**B**) Levels of intracellular cccDNA and extracellular HBsAg at day 9 after HBV infection. HepG2-hNTCP-C4 cells were treated with 10 μM ivermectin for 48 h. siRNAs targeting KPNA1–6 and KPNB1 (20 nM) were transfected at cell seeding. NT, non-transfection. The cccDNA level of the NT group was defined as 1. (**C**) Extracellular HBsAg levels at day 9 after HBV infection. HepG2-hNTCP-C4 cells were treated with 10 μM ivermectin or 10 μM importazole for 48 h. Data are presented as the mean ± SD [(**A**), *n* = 3; (**B**), *n* = 4]. * *p* < 0.05, ** *p* < 0.01 ((**A**,**B**): vs. siRNA control).

**Table 1 viruses-14-02468-t001:** PCR primer and probe sequences.

**Gene**	**Forward Primer Sequence (5′ → 3′)**	**Reverse Primer Sequence (5′ → 3′)**
*HPRT*	GCGATGTCAATAGGACTCCAG	TTGTTGTAGGATATGCCCTTGA
*KPNA1*	GTGTCTGAATATCATCCCCTGTG	GAGACTTGTGGAACTGCTGAT
*KPNA2*	ATCAACAGACCAATTTACAGTGC	AAGGATGACCAGATGCTGAAG
*KPNA3*	CTGTTCATTACCTCCATCTGTCA	GCCAGCTTTATGTGTCCTCAT
*KPNA4*	CCTTAATTCAACTACAACTTCATTTCG	CCAACGGCTCAAGAATTTCAAG
*KPNA5*	TGATTGAAACTGGGGCTGT	GCATTCTGCATTGTCACCAG
*KPNA6*	TCCATTACAGTCAGCAAGTCAC	CATCACCAATGCCACATCAG
*KPNA7*	CCAGCTCAGAACTCAATGTCT	AGCACGTTCAGCATACCC
*KPNB1*	ATTCTGCCTACAGTCCATGC	CCAGTCAGCTCAAACCACTA
**Gene**	**Probe Sequence (5′ → 3′)**	
*HPRT*	56-FAM/AGCCTAAGA/ZEN/TGAGAGTTCAAGTTGAGTTTGG/31ABkFQ	
*KPNA1*	56-FAM/TTTCTCCTG/ZEN/CTTTGCGAGCTGTG/31ABkFQ	
*KPNA2*	56-FAM/TGATGATGC/ZEN/TACTTCTCCGCTGCAG/31ABkFQ	
*KPNA3*	56-FAM/ACAGAGCCC/ZEN/AAACAGTGTCTACAAGAATG/31ABkFQ	
*KPNA4*	56-FAM/TTCTCATAG/ZEN/TCTCCAAGTCGCGGC/31ABkFQ	
*KPNA5*	56-FAM/CAGGCTGTT/ZEN/TGGGCACTTGGTAAT/31ABkFQ	
*KPNA6*	56-FAM/TGATGCAGC/ZEN/CCAGTGAGACCAG/31ABkFQ	
*KPNA7*	56-FAM/ACAGATGAG/ZEN/CAGACGCAGATGGC/31ABkFQ	
*KPNB1*	56-FAM/TGCCCACCC/ZEN/TAATAGAATTAATGAAAGACCC/31ABkFQ	

**Table 2 viruses-14-02468-t002:** Primary antibodies for Western blotting.

Antibody Name	Source	Catalog Number	Biological Source	Dilution
KPNA1	Proteintech	18137-1-AP	Rabbit	1:1000
KPNA2	GeneTex	GTX106323	Rabbit	1:1000
KPNA3	Bethyl Laboratories	A301-626A	Rabbit	1:500
KPNA4	Proteintech	12463-1-AP	Rabbit	1:2000
KPNA5	GeneTex	GTX112203	Rabbit	1:2000
KPNA6	Proteintech	12366-2-AP	Rabbit	1:1000
KPNB1	Proteintech	10077-1-AP	Rabbit	1:3000
Lamin B1	Proteintech	12987-1-AP	Rabbit	1:2000

## Data Availability

Not applicable.

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
