# Peer review of "Ivermectin Inhibits HBV Entry into the Nucleus by Suppressing KPNA2"

_viruses, 2022, doi:10.3390/v14112468_

Round 1

Reviewer 1 Report

The authors present that the compound IVM acts as an inhibitor for HBV infection. By analyzing different steps in the HBV replication cycle the authors suggest that the mode of action is the inhibition of nuclear import of HBV capsids. The Introduction is well written. For the other parts I would suggest the following edits:

Material and Methods:

Please cite the origin of the HepAD38.7-Tet cells are these the original from Ladner et al.?

Please call the “PXB cells” “PXB hepatocytes” instead. Especially in the result part it might not be clear for the reader that PXB cells are hepatocytes derived from a mouse with a humanized liver.

Please provide the sequence or a citation for the TAMRA-labeled preS1 peptide used for entry blockage.

Results:

Fig. 2 the authors should show the cytotoxic effect on Ivermectin on PXB hepatocytes using a cell viability assay according to Fig. 1. since there might be a different effect especially on primary hepatocytes compared to hepatoma cells.

Fig. 5 the authors state that IVM inhibits Importin a, however besides KPNA2 all other results are not significant. I would suggest to rephrase especially the figure description to soften the statement since the result is not significant.

Fig. 6 Importins play a critical role in cell life cycles. The authors should include cytotoxicity data compared to HBV parameters. Moreover, how do the authors explain that the control siRNA leads to a significant increase in cccDNA? Compared to the non-transfected cells there is no reduction.

Discussion:

The authors should rephrase the sentence that this report “clarify the detailed mechanism” since the reported mechanism might not be the whole mechanism and is not significant.

Conclusion:

The authors should focus the conclusion on IVM since this is their primary finding and not on the results of the siRNA knockdown.

Reviewer 2 Report

Nakanishi et al., demonstrate the anti-HBV activity of Ivermectin, an FDA approved antiparasitic drug.  The authors demonstrate that ivermectin inhibits HBsAg, the nuclear localization of HBcAg and extracellular HBV DNA. Importantly, the findings reported suggest that ivermectin is also able to reduce the levels of HBV-ccc DNA by inhibiting KPNA2 leading to to reduced nuclear localization of HBcAg.    The authors also report that ivermectin is not a polymerase inhibitor nor a HBV entry inhibitor.   Inhibition of KPNA1-6 by siRNA leads to reduction in HBV-ccc DNA levels.  The paper is well written and appropriate controls are used.  Comments and suggestions that may improve the manuscript are appended below:

Major comments:

1)      The authors demonstrate that ivermectin is able to inhibit KPNA2 and inhibition of KPNA2 (and other KPNAs) inhibits HBV-ccc DNA levels and secreted HBsAg levels.  However, they do not show the effect of KPNA2 inhibition on the nuclear localization of HBcAg.  The authors may consider this experiment as it may provide a direct link between KPNA2 levels and the nuclear translocation of HBcAg. 

2)      It may also be interesting to study the impact of ivermectin in a HBV infection model over expressing KPNA2. If ivermectin does not have anti-HBV activity when excess KPNA2 is available, it can strengthen the proposed mechanism of ivermectin-mediated inhibition of HBV replication. 

Minor comments:

1)      “HepAD38.7-Tet cells, in which HBV replication can be regulated by tetracycline, and HepG2.2.15.7 cells, which are HBV-replicating cells” – the authors may want to rephrase “HBV-replicating cells”

2)      Figure 1: Concentration of Ivermectin used should be specified in the figure legend (currently it is specified only for Figure 1F and 1G).

3)      The authors may include the concentrations of ivermectin used in all the figure legends.

4)      All figure legends should indicate the cell type used.  For example, the cells used for Figure 6A and 6B are not mentioned in the legend.

Reviewer 3 Report

HBV specifically infects human hepatocytes and increases the risks of cirrhosis and liver cancer. Nucleic acid analogs and peg-interferon (PEG-IFN) are used to treat HBV-infected patients, but can hardly cure. Novel therapeutic drugs targeting different step(s) of HBV life cycle are needed. In Nakanishi et al.’s study, Ivermectin was demonstrated to inhibit HBV entry into the nucleus by suppressing KPNA2. Mechanically, Ivermectin reduced the nuclear KPNA2 level and impaired the nuclear import of HBc. Meanwhile, KPNB1 depletion increased cccDNA level and decreased HBsAg production, suggesting HBV is transported into the nucleus without importin β. Overall, the experimental design of this study is not rigorous enough and some conclusions are not supported by sufficient evidence. Some existing studies on the nuclear import of HBV and importin α/β should be referenced.

Comments

Major:

1.     In Figure 3B, the preS1 binding assay cannot rule out the potential effect of Ivermectin on HBV entry. At least the preS1 uptake assay should be performed. (ref. PMID: 30952782)

2.     Ref.42 suggested that Ivermectin’s mode of action is likely to be through binding to the NLS-binding pocket of importin α. Hence, in Figure 5B, the detection of nuclear KPNA1-7 is not an effective method to determine the target of Ivermectin.

3.     Importin α/β was demonstrated to participate in the nuclear import of HBV nucleocapsids (PMID: 12909718 and PMID: 19864387). Transfection of siRNA targeting Importin β mRNA or expression of a dominant negative Importin β in a stable cell line supporting HBV replication resulted in the accumulation of DP-rcDNA in cytoplasm and reduction of nuclear DP-rcDNA and cccDNA (PMID: 19864387). But in this study, knockdown of KPNB1 expression increased cccDNA level (Figure 6A). Why?

4.     HBV nucleocapsids contains HBV genome and mediate the nuclear import of HBV genome. Hence, the detection of nuclear import of HBc was not appropriate in Figure 4. The nuclear import of HBV capsid in HBV infection model (HBV infects HepG2-hNTCP-C4) or HBV stable replication cell line (HepAD38.7-Tet cells) should be detected.

Minor:

1.     In Figure 1, if Ivermectin suppressed the nuclear import of HBV, why did not the HBV DNA and HBsAg decrease at 3 dpi? The same question in Figure 2B and C.

2.     In Figure 4B, the markers indicating the nucleo-plasmic separation should be detected. The same question in Figure 5B.

3.     In result 3.6, “depletion of KPNA3 and KPNB1 significantly increased HBsAg and cccDNA levels, respectively.” was wrong. Depletion of KPNB1 significantly reduced HBsAg level.

4.     In page 2, it is not accurate to cite the ref.[31], which did not demonstrate that the HBV nucleocapsids were transported into the nucleus by importin α/β. The same question to the ref.[32] in page 14.

5.     In Figure 6, the effect of knockdown KPNA1-6 and KPNB1 on the nuclear import of HBc should be detected.

Round 2

Reviewer 3 Report

The authors have addressed most of my concerns.